# Impacts of Brexit on fruit and vegetable intake and cardiovascular disease in England: a modelling study

Paraskevi Seferidi,[1] Anthony A Laverty,[1] Jonathan Pearson-Stuttard,[1,2]
Piotr Bandosz,[2,3] Brendan Collins,[2] Maria Guzman-Castillo,[2] Simon Capewell,[2]
Martin O'Flaherty,[2] Christopher Millett[1]

[1]Public Health Policy Evaluation Unit, School of Public Health, Imperial College London, London, UK
[2]Department of Public Health & Policy, University of Liverpool, Liverpool, UK
[3]Department of Preventive Medicine and Education, Medical University of Gdańsk, Gdańsk, Poland

**Correspondence to**
Paraskevi Seferidi;
paraskevi.seferidi14@imperial.ac.uk

## ABSTRACT

**Objectives** To estimate the potential impacts of different Brexit trade policy scenarios on the price and intake of fruits and vegetables (F&V) and consequent cardiovascular disease (CVD) deaths in England between 2021 and 2030.

**Design** Economic and epidemiological modelling study with probabilistic sensitivity analysis.

**Setting** The model combined publicly available data on F&V trade, published estimates of UK-specific price elasticities, national survey data on F&V intake, estimates on the relationship between F&V intake and CVD from published meta-analyses and CVD mortality projections for 2021–2030.

**Participants** English adults aged 25 years and older.

**Interventions** We modelled four potential post-Brexit trade scenarios: (1) free trading agreement with the EU and maintaining half of non-EU free trade partners; (2) free trading agreement with the EU but no trade deal with any non-EU countries; (3) no-deal Brexit; and (4) liberalised trade regime that eliminates all import tariffs.

**Outcome measures** Cumulative coronary heart disease and stroke deaths attributed to the different Brexit scenarios modelled between 2021 and 2030.

**Results** Under all Brexit scenarios modelled, prices of F&V would increase, especially for those highly dependent on imports. This would decrease intake of F&V between 2.5% (95% uncertainty interval: 1.9% to 3.1%) and 11.4% (9.5% to 14.2%) under the different scenarios. Our model suggests that a no-deal Brexit scenario would be the most harmful, generating approximately 12 400 (6690 to 23 390) extra CVD deaths between 2021 and 2030, whereas establishing a free trading agreement with the EU would have a lower impact on mortality, contributing approximately 5740 (2860 to 11 910) extra CVD deaths.

**Conclusions** Trade policy under all modelled Brexit scenarios could increase price and decrease intake of F&V, generating substantial additional CVD mortality in England. The UK government should consider the population health implications of Brexit trade policy options, including changes to food systems.

## INTRODUCTION

Trade policy exerts a powerful influence on risk factors for non-communicable disease and thus for population health.[1] One prominent mechanism is the effect of trade on food

### Strengths and limitations of this study

► This study uses nationally representative dietary intake data, robust mortality projections and both own-price and cross-price effect estimates to evaluate the impact of different Brexit scenarios on the price and intake of disaggregated fruit and vegetable subgroups and associated CVD outcomes.

► Fruit and vegetable imports were approximated using the latest available data, assuming import shares will not change after Brexit.

► Trade transaction costs due to technical regulations are based on published estimates that might also reflect costs attributed to demand changes, thus overestimating real transaction costs.

► Response to price increases of fruits and vegetables were estimated using historical purchase data and might not reflect consumer response to price change post-Brexit.

► The scenarios modelled are not exhaustive and more trade policy options might arise as Brexit negotiations continue.

environments, through changes in the availability and affordability of food commodities.[2] Evidence suggests that entering new trading blocs and opening domestic markets to international trade has been associated with increased imports, sales and intake of highly processed foods.[3–6] However, there is a lack of evidence on the effect of trade on healthy food commodities, while no previous studies have investigated the potential impacts of exiting a large trading bloc.

Brexit—the impending exit of the UK from the European Union (EU)—will have profound implications across a range of sectors. One of the main changes will involve the introduction of a new trade regime that will redefine the trade relationships between the UK and the EU and non-EU countries. This is likely to affect food environments in the UK, given its high dependency on imports to meet its dietary needs.[7] This is particularly

the case for fruits and vegetables (F&V). For example, in 2017, 84% of fruits and 43% of vegetables in the UK were imported.[8] Thus, changes to the UK trade regime are likely to affect prices of F&V by increasing costs of trade. With price being one of the main determinants of consumer behaviour in the UK,[9] Brexit could affect F&V intake with significant implications for UK diets.

Low F&V intake is a major risk factor of many non-communicable diseases, including cardiovascular disease (CVD). Approximately 1.3 million CVD deaths could have been prevented globally in 2013 if intake of F&V was higher than 500 grams/day (g/day).[10] The UK performs poorly in terms of F&V intake; only 27% of adults aged 19–64 years and 35% of those above 65 years achieve daily recommended intakes.[11] Thus, Brexit trade scenarios that change the price and availability of F&V are likely to have substantial impacts on CVD outcomes in the UK. This analysis aimed to quantify the potential impacts of F&V price changes due to Brexit on CVD deaths in England between 2021 and 2030.

## METHODS

We extended the previously validated IMPACT Food Policy model[12–14] to estimate the effect of different Brexit scenarios on the price and intake of F&V in England. We then modelled consequent impacts on cumulative coronary heart disease (CHD) and stroke mortality in English adults aged 25 years and above, stratified by age and sex, over a 10-year period in order to capture medium-term effects.

### Data sources

We used import and tariff data from the World Trade Organization (WTO)[15] and the HM Revenues and Customs[16] to estimate the effect of different Brexit scenarios on price of F&V. We describe these data in more detail in online supplementary appendix A.

The effect of price changes on intake of F&V products was estimated using published UK-specific price elasticities.[17] Price elasticities measure the change in demand of a good as a response to a change in its own price (own-price elasticity) or the price of another good (cross-price elasticity). Price elasticities for five fruit (apples and pears; bananas; citrus fruit; grapes; and other fruits) and seven vegetable (brassica, such as cabbages, cauliflower and broccoli; legumes, such as beans and peas; lettuce; onions; other vegetables; root vegetables; tomatoes) subgroups were available. We used conditional uncompensated own-price and cross-price elasticities to take into account the differential effect of Brexit on the prices of different F&V. Conditional price elasticities, usually recommended at this level of product disaggregation, assume that food expenditure available to all other products remains constant, and uncompensated price elasticities assume that consumer income is held constant.[17]

We estimated total F&V intake in English adults by sex and age (25–44, 45–74, 75+ years) using data from the National Diet and Nutrition Survey Rolling Programme (2008/2009 to 2015/2016).[18] This is a nationally representative survey of UK adults and children that uses a 4-day food diary to evaluate dietary intake. As the effect of the different Brexit scenarios varies across different F&V products, we used household purchase data (in grams/person/week) for England from the Family Food Module of the Living Costs and Food Survey 2016/2017[19] to approximate intake of F&V subgroups. The survey uses a 2-week diary to identify purchases and expenditures of foods in UK households. We adjusted the overall F&V intake using the ratio of purchases of F&V subgroups with overall F&V purchases. We assumed a constant ratio across all sex and age groups.

Changes in F&V intake were translated into mortality changes using relative risk estimates of CHD, ischaemic stroke and haemorrhagic stroke with fruit intake and vegetable intake, derived from meta-analyses of longitudinal studies and further adjusted for an effect modification by age.[20] Employing age-specific relative risks allowed us to incorporate the declining effect of age on the relationships between risk factors and CVD, which has been previously described.[21] The authors assumed a linear relationship between risk and mortality of the outcomes, that is, associations with incident CVD and CVD mortality were used interchangeably. The relative risk estimates are presented in online supplementary table 1.

Finally, this model employed sex and age (10-year groups from 25 to 34 years until 85+ years) specific CVD mortality projections between 2021 and 2030, which were estimated using a Bayesian Age-Period-Cohort model and historic CVD mortality and population data from the Office for National Statistics. The methodology is described in more detail in online supplementary appendix B.

The data and data sources used in this model are summarised in table 1.

### Modelled scenarios

The UK, as an EU Member State, is part of the EU Single Market and Customs Union. As a result, trade between the UK and the EU is tariff free and frictionless with no tariff or non-tariff trade barriers in place.[22] Trade between the UK and non-EU countries is also regulated by the EU, with common preferential trade agreements and external tariffs across all EU Member States. At an international level, trade is regulated by the WTO, which is used as a platform for its members to negotiate and ratify trade agreements.

Following the referendum in June 2016, the UK government has announced its intentions to leave the EU Single Market and Customs Union and negotiate a new trading relationship with the EU and third countries.[23] The current intention from both the UK and the EU is that this will happen 2 years after Brexit, following a transition period that will maintain the trade status quo while allowing the UK to negotiate a future permanent trading position.

**Table 1** Model inputs and data sources

| Data | Data source |
|---|---|
| Import and tariff data | World Trade Organisation[15] and HM Revenues and Customs.[16] |
| Supply data (domestic production, imports and exports) | Department for Environment, Food and Rural Affairs.[26] |
| Price elasticities (mean of annual elasticity estimates between 2000 and 2009) | Tiffin et al[17] |
| Means of F&V intake by age and sex | National Diet and Nutrition Survey Rolling Programme Years 1–8 (2008/2009 to 2015/2016).[18] |
| F&V subgroups gradient (ratio to overall F&V intake) | Purchase data from the Living Costs and Food Survey 2016/2017.[19] |
| Relative risk for CHD/ischaemic stroke/haemorrhagic stroke by serving of fruit/vegetable intake by age | Micha et al[20] |
| CHD and stroke mortality projections for England by age and sex (2021–2030) | Own estimations using historic mortality data and population projections from the Office for National Statistics. |
| Ischaemic to haemorrhagic stroke ratio, estimated using number of deaths by International Classification of Diseases (ICD) code, age, and sex in 2016 | Office for National Statistics. |

CHD, coronary heart disease; F&V, fruits and vegetables.

For this analysis, we modelled four post-Brexit trade scenarios commencing in 2021 when the transition period will end:

1. Free trading agreement with the EU and third countries (S1).
   In this scenario, the UK establishes a zero-tariff free trading agreement with the EU, while it also maintains half of its non-EU tariff-free F&V importers.
2. Free trading agreement with the EU (S2).
   Under this scenario, the UK establishes a zero-tariff free trading agreement with the EU but loses all preferential access to markets of non-EU F&V importers.
3. No-deal Brexit (S3).
   Under a no-deal scenario, the UK falls back into a WTO default position, having no preferential arrangements with the EU or non-EU importers.
4. Liberalised trade regime (S4).
   Under a liberalised trade regime, the UK trades under WTO regulations with no specific trade arrangements in place but eliminates all its F&V import tariffs. This

is an extreme case scenario that shows the impact of non-tariff trade barriers on F&V trade.

Under each scenario, we assumed two main drivers of F&V import price change. First, we expect an increase in transaction costs when trading with the EU due to a rise in post-Brexit border controls. We assumed a 5% increase in transaction costs due to rules of origin checks[24] in the scenarios that a free trading agreement is applied and an additional 4.5% increase due to checks for technical regulations in the scenarios that no preferential trade arrangements are in place.[25] We describe these estimates and its sources in detail in online supplementary appendix C. Second, we assumed changes in F&V import tariffs[15] that vary under the different Brexit scenarios. Specifically, import tariffs were assumed to be zero under a free trading agreement or a liberalised regime and equal to the EU tariffs otherwise. Table 2 demonstrates the change in tariffs and transaction costs that would occur to F&V imports from the EU, non-EU countries under a preferential arrangement with the EU and other

**Table 2** Added costs to F&V imports in each Brexit scenario

| | Imports from the EU | | Duty-free imports from non-EU countries | Dutiable imports from non-EU countries |
|---|---|---|---|---|
| | **Tariffs** | **Transaction costs** | **Tariffs** | **Tariffs** |
| **No Brexit** | **No tariffs** | **None** | **No tariffs** | **EU MFN tariffs** |
| Scenario 1 | No tariffs | Rules of origin | EU MFN tariffs*50% | EU MFN tariffs |
| Scenario 2 | No tariffs | Rules of origin | EU MFN tariffs | EU MFN tariffs |
| Scenario 3 | EU MFN tariffs | Rules of origin and technical regulations | EU MFN tariffs | EU MFN tariffs |
| Scenario 4 | Zero MFN tariffs | Rules of origin and technical regulations | Zero MFN tariffs | Zero MFN tariffs |

Scenario 1: free trading agreement with the EU and third countries; scenario 2: free trading agreement with the EU; scenario 3: no-deal Brexit; Scenario 4: liberalised trade regime (see online supplementary appendix C for more details).
EU, European Union; F&V, fruits and vegetables; MFN, most favoured nation.

non-EU countries, under each Brexit scenario. Calculations for each scenario are described in more detail in online supplementary appendix C. The estimated change in price of F&V imports was translated into change in wholesale F&V price using an estimate of imports as a percentage of overall F&V supply in the UK, obtained from the Department for Environment, Food & Rural Affairs[26] (see online supplementary appendix A for more details).

### The IMPACT Food Policy model

The IMPACT Food Policy model has been previously used to estimate the effect of different policies on dietary intake and CVD in the UK[12] and elsewhere.[27] We extended the model to estimate the effect of different Brexit scenarios on prices of F&V subgroups as described above. We then translated changes in prices into changes in intake taking into account potential substitution between subgroups. We allowed the price of each F&V subgroup to change one by one until final equilibrium was reached, that is, when prices of all F&V subgroups had changed. The intake change was the difference between intake at final equilibrium and baseline. Finally, the model translated changes in intake into changes in mortality using appropriate relative risks. We assumed common relative risks for all fruit and all vegetable subgroups. The risk reduction effect of overall F&V was estimated using a cumulative risk-reduction approach.[28] Changes in mortality were expressed as number of deaths attributed to each Brexit scenario. CVD mortality was estimated as the sum of CHD, ischaemic stroke and haemorrhagic stroke. Based on recent evidence,[29] we assumed a lag time of less than a year for changes in F&V intake to impact CVD outcomes, thus lag time was not explicitly incorporated in the model. Calculations are described in more detail in online supplementary appendix D and a schematic representation of the model is shown in online supplementary figure 1.

### Uncertainty and sensitivity analyses

We performed a probabilistic sensitivity analysis to take into account uncertainty of model inputs. We employed Monte Carlo simulations to repeatedly draw random values of the inputs from their statistical distributions and run the model across multiple iterations. The estimates and their respective statistical distributions are presented in online supplementary table 2. The median and 2.5 and 97.5 percentiles of 1000 iterations were used to produce estimates and their 95% uncertainty intervals (95% UIs).

As a response to an increase in the price of F&V, consumers may shift to other F&V alternatives. We performed an additional sensitivity analysis to take into account possible substitution between fresh F&V and canned, dried and frozen F&V. The effect of the different Brexit scenarios on price of canned, dried and frozen F&V was estimated following the same methodology as the main analysis. Then, cross-price elasticities were used to identify the effect of change in the price of canned, dried and frozen F&V on fresh F&V. Finally, the effect

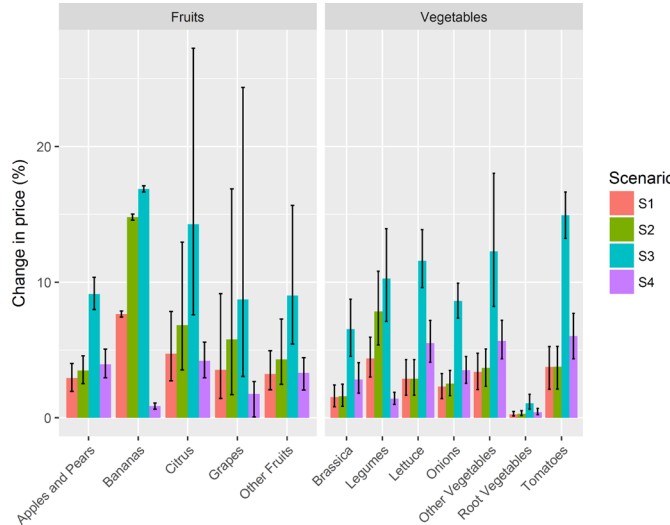

**Figure 1** Estimated relative change in price of fruit and vegetable subgroups under each modelled Brexit scenario. Error bars indicate 95% uncertainty intervals. Scenario S1: free trading agreement with the EU and third countries; scenario S2: free trading agreement with the EU; scenario S3: no-deal Brexit; scenario S4: liberalised trade regime.

of change in intake of fresh F&V on CVD mortality was calculated. As the import and purchase data used in this analysis did not allow for disaggregation of canned, dried and frozen F&V based on their content in additives, such as salt or sugar, we conservatively assumed that canned, dried and frozen F&V were not associated with CVD outcomes. Details about data sources and calculations are presented in online supplementary appendix E.

Finally, we performed a second additional sensitivity analysis that allows domestic production of F&V to increase as a response to the increase in price of imported F&V. We assumed a 2% increase per year for all F&V subgroups except bananas, citrus fruits and grapes, which have currently no domestic production in the UK.

### Patient and public involvement

There was no patient and public involvement in this study.

### RESULTS
### F&V prices

The estimated effect of Brexit scenarios on price of F&V subgroups are presented in figure 1. Price effects varied among different F&V subgroups, depending on the origin of imports, the applied tariffs and the percentage of supply that is imported.

The no-deal Brexit scenario (S3) would affect prices of F&V the most, especially for F&V that are highly dependent on imports. For example, the model suggested that under a no-deal Brexit scenario, prices would increase by 16.9% (95% UI 16.6% to 17.1%) for bananas, 14.3% (7.6% to 27.2%) for citrus fruit and 14.9% (13.2% to 16.6%) for tomatoes. In contrast, prices of root vegetables, whose imports contribute only 5% to their total supply, would increase between 0.3% (0.1% to 0.5%) and

**Table 3** Estimated relative change in intake of F&V under each modelled Brexit scenario

| Scenario | Change in intake (95% UI) | |
| | Fruits | Vegetables |
| --- | --- | --- |
| Scenario 1 | −4.4% (−5.2% to −3.8%) | −2.5% (−3.1% to −1.9%) |
| Scenario 2 | −7.0% (−8.4% to −5.9%) | −2.7% (−3.3% to −2.2%) |
| Scenario 3 | −11.4% (−14.2% to −9.5%) | −9.1% (−11.0% to −7.8%) |
| Scenario 4 | −2.8% (−3.3% to −2.3%) | −4.0% (−4.6% to −3.4%) |

Scenario 1: free trading agreement with the EU and third countries; scenario 2: free trading agreement with the EU; scenario 3: no-deal Brexit; scenario 4: liberalised trade regime.
F&V, fruits and vegetables; UI, uncertainty interval.

1.1% (0.6% to 1.7%) under different Brexit scenarios. While a liberalised trade regime (S4) will eliminate all import tariffs, it would still increase prices of F&V due to the effect of non-tariff trade barriers on EU trade. Thus, it would mostly affect prices of F&V that are primarily imported from the EU, like citrus fruits and tomatoes with an estimated 4.2% (3.0% to 5.6%) and 6.0% (4.4% to 7.7%) increase in price, respectively. More detailed price effects are presented in online supplementary table 3.

### F&V intake

We estimated that mean intake of F&V at baseline was 111 g/day and 190 g/day, respectively (online supplementary table 4). The most consumed fruits were bananas (26% of fruit intake) and apples and pears (19% of fruit intake), while root vegetables were the most consumed vegetables (17% of vegetable intake) (online supplementary table 5).

We estimated that intake of F&V would reduce under all Brexit scenarios (table 3). The no-deal Brexit scenario (S3) would have the largest effect, reducing intake of fruits by approximately 11.4% (9.5% to 14.2%) and vegetables by approximately 9.1% (7.8% to 11.0%). If a free trading agreement with the EU is established (S2), vegetable intake would reduce by 2.7% (2.2% to 3.3%), whereas intake of fruits would reduce by 7.0% (5.9% to 8.4%). Intake effects on F&V subgroups are presented in online supplementary table 6.

### CVD mortality

The number of deaths attributable to each Brexit scenario are presented in table 4 and figure 2. The no-deal Brexit scenario (S3) would be the most impactful, contributing approximately 12 400 (6690 to 23 390) extra CVD deaths or a 1.7% increase in CVD mortality over the modelling period of 2021–2030. Establishing a free trading agreement with the EU (S2) would have lower impacts on mortality contributing approximately 5740 (2860 to 11 910) extra CVD deaths or a 0.8% increase in CVD mortality over the 10-year period. Free trading agreements with both the EU and half of the UK's F&V importers (S1) would have more attenuated impacts, contributing approximately 4110 (2130 to 8100) extra CVD deaths or a 0.6% increase in CVD mortality. Even under a liberalised regime that eliminates all import tariffs (S4), Brexit would contribute approximately 4160 (2330 to 7380) extra CVD deaths or a 0.6% increase in CVD mortality between 2021 and 2030 due to an increase in transaction costs of EU imports. Online supplementary table 7 provides the estimated effects on CVD mortality for a single year of the modelling analysis.

### Sensitivity analyses

Our model suggested that prices of canned, dried and frozen F&V would increase by approximately 10.3% (5.7% to 20.0%) for fruits and 6.8% (3.9% to 16.7%) for vegetables under a no-deal Brexit scenario (S3), as shown in online supplementary table 8. Incorporating cross-price elasticities between fresh and canned, dried and frozen F&V in the model only slightly increased the effect of Brexit scenarios on intake of fresh F&V by up to 0.5% (online supplementary table 9). This would result to approximately 12 990 (7000 to 24 790) extra CVD deaths between 2021 and 2030 under a no-deal Brexit scenario (S3) (online supplementary table 10).

Our model that examined potential increases in the domestic production of some F&V did not substantially change our findings. The estimated effect on intake was 0.7% lower for fruits and 1.5% for vegetables in the last year of the modelling period compared with the first year, under a no-deal Brexit scenario (S3) (online supplementary table 11). Over the study period, this would contribute to between 210 and 720 fewer CVD deaths

**Table 4** Estimated number of cumulative CVD deaths and 95% UI for 2021–2030 associated with each modelled Brexit scenario

| Scenario | Coronary heart disease | Stroke | Cardiovascular disease |
| --- | --- | --- | --- |
| Scenario 1 | 1360 (730 to 2670) | 2740 (1400 to 5430) | 4110 (2130 to 8100) |
| Scenario 2 | 1930 (980 to 3990) | 3810 (1880 to 7910) | 5740 (2860 to 11 910) |
| Scenario 3 | 4110 (2330 to 7660) | 8290 (4360 to 15 730) | 12 400 (6690 to 23 390) |
| Scenario 4 | 1360 (790 to 2300) | 2810 (1540 to 5080) | 4160 (2330 to 7380) |

Scenario 1: free trading agreement with the EU and third countries; scenario 2: free trading agreement with the EU; scenario 3: no-deal Brexit; Scenario 4: liberalised trade regime.
CVD, cardiovascular disease; UI, uncertainty interval.

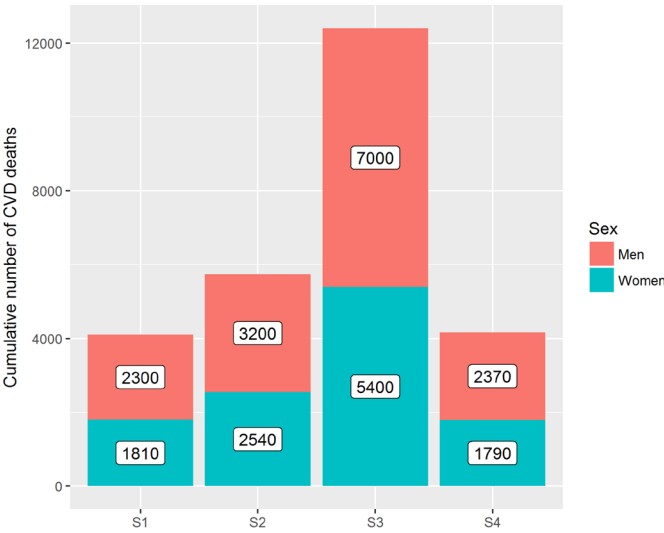

**Figure 2** Estimated number of cumulative CVD deaths for 2021–2030 associated with each modelled Brexit scenario by sex. Scenario S1: free trading agreement with the EU and third countries; scenario S2: free trading agreement with the EU; scenario S3: no-deal Brexit; scenario S4: liberalised trade regime. CVD, cardiovascular disease.

under the different scenarios compared with the main analysis for which domestic production was assumed to be stable (online supplementary table 10).

## DISCUSSION

This is the first study to exclusively quantify the potential impacts of post-Brexit trade scenarios on F&V prices, intake and consequent CVD outcomes. Estimates from all modelled scenarios suggested that F&V prices would increase, hence reducing intake and contributing to additional CVD mortality. A no-deal Brexit scenario could be the most harmful, increasing CHD and stroke deaths by approximately 0.9% (4110 deaths) and 2.9% (8290 deaths), respectively, between 2021 and 2030. The least disruptive scenario modelled, which assumes a free trading agreement with the EU and half of non-EU F&V importers, could increase CHD and stroke deaths by approximately 0.3% (1360 deaths) and 1.0% (2740 deaths), respectively.

Our findings are consistent with previous preliminary analyses that suggest likely increases in F&V prices post-Brexit. A report from the House of Lords notes the vulnerability of F&V to price changes post-Brexit.[30] An empirical estimation showed that the cost of consuming five F&V per day as per governmental guidelines will increase by £2.20 a week for a household of four under a no-deal Brexit scenario, after considering price inflation and an increase in import tariffs.[31] Analysis using a partial equilibrium model, which uses supply-and-demand curves to examine the effect of policy actions on single markets, has also shown that a no-deal Brexit might increase prices of fruits by 3.1% and vegetables by 4.0% although disaggregated effects on F&V subgroups were not further investigated.[32] These analyses did not compare a no-deal Brexit scenario with other potential post-Brexit trade options. Furthermore, they did not account for the potential effect of non-tariff trade barriers that are likely to be an important driver of price change post-Brexit.

This study has several strengths. It estimates the effect of different Brexit scenarios on the price of disaggregated F&V subgroups, taking into account both own-price and cross-price effects to estimate changes in F&V intake. Sensitivity analysis also incorporated the additional effects of substitution with canned, dried and frozen F&V. This study uses nationally representative data on F&V intake and purchases. It also employs CVD mortality projections using a Bayesian age-period-cohort model, which has been shown to produce more robust predictions compared with conventional methods.[33]

However, it also has some limitations. We used the most recent import data from 2015 to approximate F&V imports post-Brexit, assuming that F&V import shares will not change. This assumption might overestimate the price effects of scenarios that involve free trading agreements (S1 and S2) as the UK could shift towards importers that trade under these free trading agreements. There might be some discrepancies in the definition of F&V subgroups across different datasets. We explicitly describe each definition in online supplementary appendix A. Trade transaction costs due to technical regulations are based on published estimates derived using non-tariff measure data from 65 countries.[25] These estimates might overestimate the real costs as they might also reflect a price effect attributed to an increase in demand after complying with quality improving technical regulations in some countries. In this analysis, we conservatively omitted the potential impact of inflation on F&V price to minimise model uncertainty, which might underestimate price effects. We used robust estimates of UK specific price elasticities.[17] However, these elasticities were estimated using historical purchase data (2000–2009) and might not reflect the consumer response to price change after a potential wider economic shock of Brexit. Our model assumes a linear association between price, intake and CVD outcomes under the different Brexit scenarios. However, real-life associations in the medium-term and long-term may be characterised by feedback mechanisms in response to system disruptions, such as Brexit. This can be addressed using computational general equilibrium models, which have been widely used to estimate Brexit effects on trade, productivity and welfare[34 35] but are more appropriate when modelling wider economic impacts across various sectors. Although this study includes a sensitivity analysis that investigates a potential increase in domestic production of F&V as a feedback response to an increase in price of imports, it does not provide a comprehensive investigation of the economic feedback effects of our modelled Brexit scenarios. Finally, we modelled four post-Brexit trade scenarios reflecting governmental and expert consensus.[36] However, ongoing negotiations within the British government and the Conservative party[37] and

between the British government and the EU[38] means that the political space around Brexit is continuously changing and more trade policy options might arise.

F&V intake in the UK is suboptimal with less than half of the population consuming daily recommended levels.[11] Our analysis suggests that all modelled Brexit trade policy scenarios could increase price and reduce intake of F&V. The UK government needs to develop a comprehensive policy response against this potential detrimental impact by aligning domestic nutrition and agricultural policies and future trade policy with public health aims. At the same time, the wider implications of Brexit to the UK food system should also be considered. For example, approximately 80 000 seasonal workers are employed in the UK every year to harvest F&V, with the majority of them recruited from EU countries.[39] Abolishment of the free movement of labour guaranteed by EU regulations is likely to reduce available workforce for the horticulture sector with potential implications to prices of domestically produced F&V. Brexit may also affect food quality and safety in the UK by potentially relaxing risk prevention requirements, like the precautionary principle for food, which is currently eliminated from the EU (Withdrawal) Act 2018,[40] in order to increase competitiveness in international markets. Moreover, Brexit is likely to affect prices and intake of other food groups with potential implications for health. For example, a previous analysis has shown that a no-deal Brexit scenario would increase prices of all food products, with higher impacts on dairy, oils and fats and meat.[32] Finally, the wider effects of Brexit on the UK economy could affect the whole food supply chain, including packaging, distribution and retail of both fresh and processed foods. Future work should monitor and evaluate the population health implications of these potentially important changes in the UK food system.

In summary, post-Brexit trade policy could increase price and decrease intake of F&V, thus increasing CVD mortality in England. The UK government should therefore carefully consider the population health implications of Brexit during upcoming negotiations and post-Brexit planning, particularly adverse changes to food systems.

**Contributors** PS, AAL, JP-S, MO and CM conceived the study. PS and PB analysed the data with inputs from JP-S, BC, MG-C, AAL and MO. PS drafted the paper with inputs from AAL, CM, JP-S, BC, MG-C, SC and MOF. All authors made a substantial contribution to the data interpretation and critical review of the submitted manuscript.

**Funding** PS, AAL and CM are funded by the National Institute for Health Research (NIHR) via a Research Professorship Award to CM (grant number: RP 2014-04-032). NIHR had no role in the design, analysis or writing of this article. The Public Health Policy Evaluation Unit is grateful for the support of the NIHR School of Public Health Research.

**Competing interests** None declared.

**Patient consent for publication** Not required.

**Provenance and peer review** Not commissioned; externally peer reviewed.

**Data sharing statement** Model input data are available on request from the corresponding author.

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
