## [Reviewer comments · BMJ Open]

ARTICLE DETAILS

TITLE (PROVISIONAL)	Impacts of Brexit on fruit and vegetable intake and cardiovascular disease in England: a modelling study
AUTHORS	Seferidi, Paraskevi; Lavery, Anthony; Pearson-Stuttard, Jonny; Bandosz, Piotr; Collins, Brendan; Castillo, Maria; Capewell, Simon; O'Flaherty, M; Millett, Christopher

VERSION 1 – REVIEW

REVIEWER	Marco Springmann University of Oxford, UK
REVIEW RETURNED	16-Nov-2018

GENERAL COMMENTS	In the manuscript “Impacts of Brexit on fruit and vegetable intake and cardiovascular disease in England: a modelling study,” Paraskevi Seferidi and colleagues aim to estimate the potential impacts that different Brexit scenarios could have on the price and intake of fruits and vegetables in the UK, including the associated impacts on cardiovascular disease mortality. The topic is of clear interest to the public, public health professionals, and policy-makers. The manuscript is well written, and the methods are comprehensively documented. However, the analysis would benefit from some additional contextualisation. The focus of the study is on how fruit and vegetable intake could change due to Brexit. Whilst that is clearly important for public health, fruits and vegetables are not the only risk factors in our diets. For example, red meat consumption could decline as well as a high proportion of meat products are currently imported. And with most foods being subject to new tariffs, one could even expect a small decline in total energy intake. Both of those impacts would be associated with health impacts. I do think the focus on fruits and vegetables is a good and important one, but I would suggest to at least mention that Brexit can be expected to also have an impact on other dietary risk factors. Also the economic analysis would benefit from added context. Methodologically, the line of reasoning in the study goes from changes in trade costs to changes in prices to changes in consumption to changes in disease risk to changes in mortality. As such, this is a reasonable succession of processes, but the step from changes in prices of a small group of commodities to changes in consumption of that group of commodities describe short-term impacts only. In multi-year analysis, what one would expect are economic feedback effects that attenuate those short-term impacts, for example by reducing exports or increasing production. The sensitivity analysis goes some way in testing the implications of an arbitrarily assumed increase in production, but that is not exactly a
---

consistent economic analysis. That said, I still think that the analysis done is useful and provides insights into the potential first-order effects of Brexit, but I would expect a bit more methodological contextualisation (beyond mentioning that import shares were assumed not to change after Brexit).

Here I should probably mention that I and a colleague of mine recently did an analysis of the potential health impacts of Brexit using a class of models that is more suited to study the economic feedback effects in the medium term (published here as a preprint, something that is common for economic papers: <https://www.oxfordmartin.ox.ac.uk/publications/view/2754>). In our economic analysis, we used a computable general equilibrium (CGE) model with added agricultural detail. In general, CGE models provide a structural representation of a country's or region's economy by combine economic theory with empirical economic data. Due to their structure, such models allow for the analysis of economic feedback effects throughout multiple economic sectors and regions that go beyond the first order effects captured by the analysis under review here. Because of those features, CGE models have become the model of choice in the branch of applied economic research that is focused on trade analysis. Many CGE analyses have been devoted to Brexit (see for example the overview by Sampson, doi:10.1257/jep.31.4.163), so I was surprised to not see that important body of Brexit-related literature mentioned in the manuscript.

Where our analysis is more detailed and consistent on the economic end, the analysis under review here has greater commodity detail, so I would see the two analyses as complementary. It also might be good to note for judging the validity of the present study that we find that the output effects for fruits and vegetables are relatively minor (an increase of 1.4% over ten years under a soft Brexit and a slight decline of 0.2% under a hard Brexit). Among other things, this is the result of factors of production directed towards the production of pork and poultry which are sectors that would face greater increases in trade costs under Brexit and which therefore have a greater incentive to increase production. But enough about our study now.

Coming to the health model used in the analysis under review, I have two short questions. Did you assume any time lag between disease incidence and disease mortality. If not, then you might want to mention that as a limitation in the analysis. That would seem important if a longitudinal model of disease mortality was used. (Of course, applying a longitudinal model to short-term consumption changes is slightly inconsistent as well, so you might want to caveat that in some way as well, or mention the relatively modest output changes noted above.) Second, as source for your relative risk estimates you are referring to an application of such risk parameters (Micha et al, 2017) instead of an actual meta-analysis of cohort studies which would be a more appropriate source. To my knowledge, the most recent meta-analysis of cohort studies focused on CVDs and fruits and vegetables was done by Aune and colleagues (doi:10.1093/ije/dyw319), and I see that you cite that paper in your manuscript. Is there a reason you didn't use their relative risk estimates for the actual analysis? I wouldn't expect the effects to be dramatically different, so I am just curious here. Also, please provide a table with the relative risk values you used.

	My last comment is about the discussion of the various Brexit scenarios that were analysed. I do think they span a nice range of potential outcomes and scenarios that have been under discussion (I also like the overview you provide in Table C1). I am just wondering if a bit more context would be beneficial here. For example, the liberalised trade scenario (scenario 4) seems pretty far fetched to me, and almost implausible to reach within a ten-year time frame (if at all), considering for example how long it took the EU to negotiate the several dozens of trade agreements with third countries. So a bit more discussion on the motivation for and representation of the different scenarios would be very welcome. Minor comments: P 4, l 39: Typo, should read 'impacts on CVD outcomes'. P 5, l 38-39: Was the purchase data provided in terms of weight? Otherwise the scaling would be biased towards high-value foods. P 8, l 24-25: I am not aware that frozen or canned fruits and vegetables are excluded from the cohort analyses that derive the relative risk estimates you would have used. Assuming no association, in particular for frozen and canned vegetables seem like a strong assumption to me. P 9, l 3-4: I find it not straightforward to understand why a liberalised trade regime would affect prices of F&V that are primarily imported from the EU. Please explain that a bit more. I presume you mean because it is only on EU trade that new non-tariff barriers are applied in that scenario? P 9, l 51: You might want to explain why you analysed CVD mortality over a ten-year time period. P 10, l 45: Well, I'm not sure who was first, but we probably did our analysis at about the same time. So you might want to change 'first study' to 'one of the first studies' or to 'first study to exclusively analyse the potential impacts of Brexit on fruits and vegetables' as we looked at other risk factors as well. P 2 in appendix, l 27: Typo, should read 'throughout the year'. P2 in the appendix, l 34: Is there any reason you averaged over the last ten years here?
--	---

REVIEWER	James Milner London School of Hygiene & Tropical Medicine, UK
REVIEW RETURNED	21-Nov-2018

GENERAL COMMENTS	This paper addresses the important and timely issue of whether the UK's withdrawal from the EU will affect consumption of fruits and vegetables and hence cardiovascular mortality in England. It is an excellent and well presented piece of work that I would recommend for publication. The health modelling is based on sound methods. However, I am not an economist and would recommend review by an expert in that field to comment more thoroughly on the economic/policy components of the model. I have relatively little to add but wondered whether a few minor tweaks would help readers to understand the modelling process. First, if possible, I think some of the information from the supplementary materials could perhaps be moved to the main text. In particular, the information in Appendix C drives the changes in price (and therefore in consumption and health). Since this information is crucial to the model, I think it would be helpful for some of the detail here to be moved to the main text.
---

	Similarly, a schematic diagram of the modelling 'chain' in the Methods section would be helpful for the reader. This diagram could show explicitly where Brexit exerts an influence on the system being modelled. Finally, I would move Table C1 to the main text since it provides a concise and helpful summary of the four scenarios. A few other minor points of clarification:  - You have assumed that import shares will not change after Brexit. This seems to be a critical assumption. Would it be possible to test the sensitivity of your model to that assumption? - In scenario 1, what is the basis for the assumption that the UK maintains half of its non-EU tariff-free importers? Again, do you know if your model is sensitive to this? - Are there other drivers of import prices that are not included in your model and, if so, how important might they be?
--	--

VERSION 1 – AUTHOR RESPONSE

Replies to Reviewers' Comments to Author:

Reviewer: 1

Reviewer Name: Marco Springmann

Institution and Country: University of Oxford, UK

Please state any competing interests or state 'None declared': I and a colleague of mine currently have a manuscript on the same topic under review in the same journal. I mention that in my comments to the authors.

In the manuscript "Impacts of Brexit on fruit and vegetable intake and cardiovascular disease in England: a modelling study," Paraskevi Seferidi and colleagues aim to estimate the potential impacts that different Brexit scenarios could have on the price and intake of fruits and vegetables in the UK, including the associated impacts on cardiovascular disease mortality. The topic is of clear interest to the public, public health professionals, and policy-makers. The manuscript is well written, and the methods are comprehensively documented. However, the analysis would benefit from some additional contextualisation.

We thank the reviewer for their positive comments and we have addressed their specific concerns below.

The focus of the study is on how fruit and vegetable intake could change due to Brexit. Whilst that is clearly important for public health, fruits and vegetables are not the only risk factors in our diets. For example, red meat consumption could decline as well as a high proportion of meat products are currently imported. And with most foods being subject to new tariffs, one could even expect a small decline in total energy intake. Both of those impacts would be associated with health impacts. I do think the focus on fruits and vegetables is a good and important one, but I would suggest to at least mention that Brexit can be expected to also have an impact on other dietary risk factors.

We agree with the reviewer that Brexit could affect prices of many agricultural products, beyond fruits and vegetables. We decided to focus on fruits and vegetables in this paper as they are the most

imported agricultural commodity in the UK and have significant impacts on health. Additionally, focusing on one food group allowed us to employ a model with higher granularity including investigating potential differential effects on prices of different fruit and vegetable subtypes. We agree with the reviewer, however, that the effect of Brexit on the overall diet beyond fruits and vegetables could be significant. As suggested, we now clarify that Brexit could increase prices of many food groups, including meat and dairy, on page 13, lines 40-47 of the manuscript.

Also the economic analysis would benefit from added context. Methodologically, the line of reasoning in the study goes from changes in trade costs to changes in prices to changes in consumption to changes in disease risk to changes in mortality. As such, this is a reasonable succession of processes, but the step from changes in prices of a small group of commodities to changes in consumption of that group of commodities describe short-term impacts only. In multi-year analysis, what one would expect are economic feedback effects that attenuate those short-term impacts, for example by reducing exports or increasing production. The sensitivity analysis goes some way in testing the implications of an arbitrarily assumed increase in production, but that is not exactly a consistent economic analysis. That said, I still think that the analysis done is useful and provides insights into the potential first-order effects of Brexit, but I would expect a bit more methodological contextualisation (beyond mentioning that import shares were assumed not to change after Brexit). Here I should probably mention that I and a colleague of mine recently did an analysis of the potential health impacts of Brexit using a class of models that is more suited to study the economic feedback effects in the medium term (published here as a preprint, something that is common for economic papers: <https://www.oxfordmartin.ox.ac.uk/publications/view/2754>). In our economic analysis, we used a computable general equilibrium (CGE) model with added agricultural detail. In general, CGE models provide a structural representation of a country's or region's economy by combine economic theory with empirical economic data. Due to their structure, such models allow for the analysis of economic feedback effects throughout multiple economic sectors and regions that go beyond the first order effects captured by the analysis under review here. Because of those features, CGE models have become the model of choice in the branch of applied economic research that is focused on trade analysis. Many CGE analyses have been devoted to Brexit (see for example the overview by Sampson, doi:10.1257/jep.31.4.163), so I was surprised to not see that important body of Brexit-related literature mentioned in the manuscript.

We thank the reviewer for these insightful comments. In this analysis, we used the IMPACT Food Policy model which is a linear comparative risk assessment model and does not incorporate feedback effects. As the reviewer noted, we have attempted to address this limitation by performing a sensitivity analysis which assumes a feedback effect on domestic production of fruits and vegetables due to the increase in price of imports. We found that even an annual 2% increase in fruit and vegetable production did not substantially alter our findings. We agree that an econometric analysis using CGE methodology, like the one used in the reviewer's paper, will address feedback effects more explicitly. We have now clarified that our model does not provide a comprehensive investigation of potential feedback effects and have addressed the advantages of CGE models in these type of studies, citing relevant Brexit-related literature as per the reviewer's suggestion (page 12 line 56 to page 13, line 12).

Where our analysis is more detailed and consistent on the economic end, the analysis under review here has greater commodity detail, so I would see the two analyses as complementary. It also might be good to note for judging the validity of the present study that we find that the output effects for fruits and vegetables are relatively minor (an increase of 1.4% over ten years under a soft Brexit and a slight decline of 0.2% under a hard Brexit). Among other things, this is the result of factors of production directed towards the production of pork and poultry which are sectors that would face greater increases in trade costs under Brexit and which therefore have a greater incentive to increase production. But enough about our study now.

Thank you for these interesting insights into your paper and we agree that the two studies are complementary. In our sensitivity analysis, we modelled an annual 2% increase in domestic production, which is higher than what you identified in your paper, but still found little impact on our estimated results.

Coming to the health model used in the analysis under review, I have two short questions. Did you assume any time lag between disease incidence and disease mortality. If not, then you might want to mention that as a limitation in the analysis. That would seem important if a longitudinal model of disease mortality was used. (Of course, applying a longitudinal model to short-term consumption changes is slightly inconsistent as well, so you might want to caveat that in some way as well, or mention the relatively modest output changes noted above.)

We thank the reviewer for these valuable comments. We assumed no lag time between disease incidence and mortality. This is because the relative risks used in this analysis assume no differential associations of fruit and vegetable intake with CVD incidence and CVD mortality, i.e. fruit and vegetable intake do not affect the fatality of the event. We have now explicitly mentioned this on page 5, lines 57-59. Moreover, the model does project secular CVD mortality rates over the study period hence it does indirectly incorporate the fact that CVD mortality is changing. In terms of lag time between changes in risk factor exposure and CVD, we assumed that it would be lower than one year, thus it was not incorporated into the model. This assumption is supported by relevant evidence now mentioned on page 8, lines 45-50.

Second, as source for your relative risk estimates you are referring to an application of such risk parameters (Micha et al, 2017) instead of an actual meta-analysis of cohort studies which would be a more appropriate source. To my knowledge, the most recent meta-analysis of cohort studies focused on CVDs and fruits and vegetables was done by Aune and colleagues (doi:10.1093/ije/dyw319), and I see that you cite that paper in your manuscript. Is there a reason you didn't use their relative risk estimates for the actual analysis? I wouldn't expect the effects to be dramatically different, so I am just curious here. Also, please provide a table with the relative risk values you used.

In this analysis, we used RR estimates from Micha et al because they were age-stratified. In their comparative risk assessment model, Micha et al used relative risks from published or de-novo meta-analyses to which they incorporated an effect size modification by age. This makes these estimates more appropriate for comparative risk assessment models stratified by age, as used in our analysis. Age-specific relative risks allowed us to incorporate the previously shown declining effect of age on the relationships between risk factors and CVD into our analysis. We now expand on this on page 5, lines 45-58 of the revised manuscript. A table with the RR estimates used is also provided in the Appendix and cited on page 5, line 59 of the manuscript.

My last comment is about the discussion of the various Brexit scenarios that were analysed. I do think they span a nice range of potential outcomes and scenarios that have been under discussion (I also like the overview you provide in Table C1). I am just wondering if a bit more context would be beneficial here. For example, the liberalised trade scenario (scenario 4) seems pretty far fetched to me, and almost implausible to reach within a ten-year time frame (if at all), considering for example how long it took the EU to negotiate the several dozens of trade agreements with third countries. So a bit more discussion on the motivation for and representation of the different scenarios would be very welcome.

We thank the reviewer for this interesting point and are pleased that they recognise our attempts to model a broad range of scenarios. We agree that the current political space indicates that a liberalised regime is not in the immediate plans of the UK government. The liberalised regime

scenario assumes that the UK chooses to apply zero Most-Favoured-Nation (MFN) tariffs on fruits and vegetables without having to negotiate specific preferential arrangements with other countries. We believe that this scenario, although not very likely, showcases the important effect of non-tariff trade barriers on trade costs. It shows that even with an elimination of import tariffs, including those currently applied by the UK as an EU Member State, Brexit could still increase prices of F&V due to increased non-tariff trade barriers on EU trade. We have now explicitly addressed this in our Methods (page 7, lines 30-32).

Minor comments:

P 4, l 39: Typo, should read 'impacts on CVD outcomes'.

Thank you, we have now corrected this.

P 5, l 38-39: Was the purchase data provided in terms of weight? Otherwise the scaling would be biased towards high-value foods.

The reviewer is correct that purchase data were indeed in terms of weight and we are now clearer about this on page 5, line 36 of the revised manuscript.

P 8, l 24-25: I am not aware that frozen or canned fruits and vegetables are excluded from the cohort analyses that derive the relative risk estimates you would have used. Assuming no association, in particular for frozen and canned vegetables seem like a strong assumption to me.

Thank you for this comment on our sensitivity analyses. The decision to separate canned, dried, and frozen fruits and vegetables from their fresh equivalents was driven by the categorisation of fruits and vegetables in the various datasets used. Both the Harmonised System-4 classification used by trade data and the Living Cost and Food Survey purchase data do not specify if canned, dried, and frozen preparations of fruits and vegetables contain added salt, sugar, and other flavourings and/or they combine them in one aggregated group. The relative risk estimates we use specify that these types of fruits and vegetables are excluded. As a result, we conservatively assumed that all frozen, dried, and canned preparations of fruits and vegetables were not associated with CVD outcomes. We have now specified this on page 9, lines 17-21 of the revised manuscript.

P 9, l 3-4: I find it not straightforward to understand why a liberalised trade regime would affect prices of F&V that are primarily imported from the EU. Please explain that a bit more. I presume you mean because it is only on EU trade that new non-tariff barriers are applied in that scenario?

Yes, indeed. Under a liberalised regime, the only drivers of price increase would be the non-tariff trade barriers. These are only going to affect EU trade which is currently frictionless contrary to trade with third countries which has already been subjected to non-tariff trade barrier costs. We have now explained that increases in prices of F&V under a liberalised regime were due to the effect of non-tariff trade barriers on EU trade on page 9, lines 55-59 of the revised manuscript.

P 9, l 51: You might want to explain why you analysed CVD mortality over a ten-year time period.

This analysis focuses on modelling the medium-term effects of Brexit, in accordance with previous literature on CVD and dietary changes. This is now specified on page 4, lines 55-57.

P 10, l 45: Well, I'm not sure who was first, but we probably did our analysis at about the same time. So you might want to change 'first study' to 'one of the first studies' or to 'first study to exclusively analyse the potential impacts of Brexit on fruits and vegetables' as we looked at other risk factors as well.

Thank you for your note, we have amended the sentence on page 11, line 48 of the manuscript.

P 2 in appendix, l 27: Typo, should read 'throughout the year'.

Thank you, typo is corrected.

P2 in the appendix, l 34: Is there any reason you averaged over the last ten years here?

We used a 10-year average to capture annual variations at the medium term. We have now clarified this on page 2, line 34 of the Appendix.

Reviewer: 2

Reviewer Name: James Milner

Institution and Country: London School of Hygiene & Tropical Medicine, UK

Please state any competing interests or state 'None declared': None declared

This paper addresses the important and timely issue of whether the UK's withdrawal from the EU will affect consumption of fruits and vegetables and hence cardiovascular mortality in England. It is an excellent and well presented piece of work that I would recommend for publication.

The health modelling is based on sound methods. However, I am not an economist and would recommend review by an expert in that field to comment more thoroughly on the economic/policy components of the model.

I have relatively little to add but wondered whether a few minor tweaks would help readers to understand the modelling process.

We thank the reviewer for their positive comments and we discussed their specific points below.

First, if possible, I think some of the information from the supplementary materials could perhaps be moved to the main text. In particular, the information in Appendix C drives the changes in price (and therefore in consumption and health). Since this information is crucial to the model, I think it would be helpful for some of the detail here to be moved to the main text.

Thank you for this recommendation. We have now expanded our methods explaining the types of tariff and non-tariff trade barriers that we assume under each scenario, as can be seen on page 7, lines 36-45 of the revised manuscript.

Similarly, a schematic diagram of the modelling 'chain' in the Methods section would be helpful for the reader. This diagram could show explicitly where Brexit exerts an influence on the system being modelled.

Thank you for this suggestion. We have now added a schematic representation of the model and its main inputs in the Appendix, page 19 and reference it in the manuscript, page 8, line 51.

Finally, I would move Table C1 to the main text since it provides a concise and helpful summary of the four scenarios.

In line with a similar suggestion from reviewer 1, we have now moved Table C1 from the Appendix to the main document on page 8, line 1 (Table 2).

A few other minor points of clarification:

- You have assumed that import shares will not change after Brexit. This seems to be a critical assumption. Would it be possible to test the sensitivity of your model to that assumption?

Thank you for this note. Indeed, our model does not take into account potential changes in import shares under the different scenarios. This assumption only affects scenarios 1 and 2 that involve free trading agreements. In scenarios 3 (no deal Brexit) and 4 (liberalised regime), trade costs are assumed to be equal for all of UK's trading partners so there would be no motivation to shift trade partners. We have now clarified that this assumption might overestimate price effects under scenarios 1 and 2 in our Discussion, page 12, lines 36-40.

- In scenario 1, what is the basis for the assumption that the UK maintains half of its non-EU tariff-free importers? Again, do you know if your model is sensitive to this?

We thank the reviewer for this comment. Scenario 1 assumes that the UK will re-establish free trading agreements with 50% of non-EU countries. This is an arbitrary percentage but we think that it provides the reader with a sufficient range of scenarios together with Scenario 2 (re-establishing free trading agreements with 0% of non-EU countries) and the baseline no Brexit scenario (maintaining 100% of free trading agreements with non-EU countries).

- Are there other drivers of import prices that are not included in your model and, if so, how important might they be?

Thank you for pointing this out. In this analysis, we did not account with the potential effects of price inflation post-Brexit. We conservatively made this decision due to the uncertainty of sterling's trajectory post-Brexit. A further depreciation of sterling post-Brexit could further increase our estimated price and CVD mortality effects. We now discuss this on page 12, lines 49-52 of the revised manuscript. Other factors, such as labour and product quality, could have indirect effect on fruit and vegetable price as discussed on page 13, lines 30-41.

VERSION 2 – REVIEW

REVIEWER	Marco Springmann University of Oxford, UK
REVIEW RETURNED	04-Jan-2019

GENERAL COMMENTS	The authors have sufficiently addressed my comments and I am happy to recommend it for publication.
---

REVIEWER	James Milner London School of Hygiene & Tropical Medicine, UK
REVIEW RETURNED	03-Jan-2019

GENERAL COMMENTS	Thank you for updating the manuscript to reflect my comments and those of the other reviewer. I am satisfied with the amendments made to the manuscript and would recommend publication.
--